# Age-dependent changes in protein incorporation into collagen-rich tissues of mice by in vivo pulsed SILAC labelling

Yoanna Ariosa-Morejon[1], Alberto Santos[2,3], Roman Fischer[4], Simon Davis[4], Philip Charles[4], Rajesh Thakker[5], Angus KT Wann[1], Tonia L Vincent[1]*

[1]Kennedy Institute of Rheumatology, Arthritis Research UK Centre for OA Pathogenesis, University of Oxford, Oxford, United Kingdom; [2]Big Data Institute, Li-Ka Shing Centre for Health Information and Discovery, University of Oxford, Oxford, United Kingdom; [3]Center for Health Data Science, Faculty of Health Sciences, University of Copenhagen, Copenhagen, United Kingdom; [4]Nuffield Department of Medicine, Target Discovery Institute, University of Oxford, Oxford, United Kingdom; [5]Academic Endocrine Unit, OCDEM, Churchill Hospital, University of Oxford, Oxford, United Kingdom

**Abstract** Collagen-rich tissues have poor reparative capacity that predisposes to common age-related disorders such as osteoporosis and osteoarthritis. We used in vivo pulsed SILAC labelling to quantify new protein incorporation into cartilage, bone, and skin of mice across the healthy life course. We report dynamic turnover of the matrisome, the proteins of the extracellular matrix, in bone and cartilage during skeletal maturation, which was markedly reduced after skeletal maturity. Comparing young adult with older adult mice, new protein incorporation was reduced in all tissues. STRING clustering revealed changes in epigenetic modulators across all tissues, a decline in chondroprotective growth factors such as FGF2 and TGFβ in cartilage, and clusters indicating mitochondrial dysregulation and reduced collagen synthesis in bone. Several pathways were implicated in age-related disease. Fewer changes were observed for skin. This methodology provides dynamic protein data at a tissue level, uncovering age-related molecular changes that may predispose to disease.

*For correspondence: tonia.vincent@kennedy.ox.ac.uk

Competing interest: The authors declare that no competing interests exist.

## Introduction

As life expectancy extends, the societal burden of age-related diseases is predicted to increase substantially. Ageing, the natural decline of cellular and physiological processes during life, is particularly apparent in collagen-rich tissues such as the articular cartilage, bone, and skin, leading to osteoarthritis (OA), osteoporosis, and impaired cutaneous wound healing in the elderly adults. Each of these tissues is characterised by an abundance of extracellular matrix (ECM), a dynamic network composed of collagens, proteoglycans, glycoproteins, and ECM-associated proteins, defined collectively as the matrisome (*Naba et al., 2016*). The matrisome can constitute the majority of the tissue by volume, up to 95% in the case of articular cartilage (*Sophia Fox et al., 2009*), and is critically important in controlling the phenotype of the cells embedded within it. The cells responsible for matrix production and homeostasis include those that are post-mitotic after skeletal maturity, such as chondrocytes in articular cartilage and the osteocytes in bone, or those that may be renewable through repopulation (osteoclasts derived from blood mononuclear cells) or through proliferation, for example fibroblasts in skin and osteoblasts in bone (although this is often blunted with age). The ability to examine synthetic

activity in these tissues in situ under normal physiological conditions, without disrupting the tissue, is highly valuable.

Several mechanisms have been proposed that account for changes to the ECM with age, including reduced synthesis, altered turnover (reduced and increased), and post-translational modifications. Of the latter, described processes include accumulation of advanced glycation end products (AGEs), deamidation and racemisation of amino acid residues, crosslinking of matrix macromolecules, and accumulation of protein aggregates (*Hsueh et al., 2019*; *Verzijl et al., 2000*; *Ewald, 2019*). These impact on the biophysical properties and regenerative capacity of the tissue and predispose to disease (*Kim et al., 2015*). In bone, the accumulation of AGEs, combined with collagen loss and cellular senescence, are regarded as critical drivers of osteoporosis (*Sanguineti et al., 2014*; *Shuster, 2005*). Loss of type II collagen turnover in mature cartilage is thought to predispose to poor regenerative capacity and OA. Similarly, reduced turnover of the matricellular proteins in dentin, tendon, and skin likely result in impaired regenerative responses and elsewhere may contribute to tissue fibrosis and cancer risk (*Neill et al., 2016*). In addition to structural and biomechanical impacts of matrisomal changes, signals from ECM protein fragments and growth factors associated with the ECM (*Hynes, 2009*) can bind directly to cell-surface receptors to modulate cell proliferation and survival, cell morphology, and tissue metabolism (*Huang and Greenspan, 2012*).

Quantitative proteomic studies provide valuable information regarding molecular composition and abundance of tissue protein, but this only represents a snapshot of the physiological state at a given time and does not accurately account for dynamic protein turnover within tissues (*Claydon et al., 2012*). A number of methodologies have been applied to investigate dynamic protein changes over time. Radiolabelling methods and methods that measure post-translational modifications (racemisation, AGE, and D-aspartate accumulation), in human skin, cartilage, dentin, and tendon, have identified major fibrillar collagens as very stable proteins with reduced incorporation after skeletal maturity and half-lives as long as 117 years in cartilage (type II collagen) and 15 years in skin (type I collagen) (*Heinemeier et al., 2016*; *Libby et al., 1964*; *Verzijl et al., 2000*; *Maroudas et al., 1998*; *Verzijl et al., 2001*). A limitation of these methods is that time is not the only factor influencing these changes. For example, high temperatures, common at injury sites, can accelerate racemisation (*Dyer et al., 1993*; *Maroudas et al., 1992*; *Stabler et al., 2009*). Metabolic labelling is seen as a more direct method to estimate protein lifespans. Incorporation of $^{14}$C using the bomb-pulse method takes advantage of an increase in atmospheric $^{14}$C that resulted from nuclear testing in the 1950 s (*Libby et al., 1964*; *Lynnerup et al., 2008*; *Nielsen et al., 2016*). This method is suitable for estimating turnover of bulk tissue proteins or select protein types but is incompatible with more agnostic approaches using mass spectrometry. Another method using deuterated water ($^{2}$H$_2$O) has been used to estimate protein synthesis and turnover rates in several animal models and tissues (*Choi et al., 2020*; *Kim et al., 2012*). With this approach, animals incorporate deuterium into C-H bonds, thus labelling newly synthesised proteins and other biomolecules. The increase in peptide mass over time indicates neosynthesis. A limitation of this method is that because $^{2}$H$_2$O is incorporated into multiple biomolecules, labelled amino acids can derive directly from the diet during the labelling period or from recycled, previously labelled, biomolecules (*Alevra et al., 2019*).

Stable isotope labelling (Lys (6)-SILAC-Mouse Diet, SILANTES GmbH) incorporated into the diet overcomes many of the aforementioned problems and has been used to determine incorporation rates of proteins from blood cells, organelles, and organs in vivo (*Kruger et al., 2008*). In this diet, the six naturally abundant $^{12}$C molecules in Lysine have been replaced with $^{13}$C, conferring a molecular mass shift of 6 Daltons for each lysine in a proteolytically cleaved peptide. This method was originally used to perform quantitative comparisons of the proteome of fully labelled (F2 generation) mice to determine protein dynamics in vivo (*Kruger et al., 2008*). Quantification of the 'heavy' and 'light' protein fractions by mass spectrometry over time provides a measure of the incorporation rates of individual proteins (*Ebner and Selbach, 2011*; *Schwanhausser et al., 2009*). In the study, we pulse label mice with SILAC diet for 3 weeks at three different post-natal periods: weeks 4–7 (maximum skeletal growth), weeks 12–15 (young adult), and weeks 42–45 (older adult), then apply mass spectrometry to quantify new protein incorporation into three collagen-rich tissues (articular cartilage, bone, and skin) and plasma, which was used as a high-turnover reference tissue, allowing us to estimate and compare new protein incorporation rates at proteome-wide scale across the healthy life course.

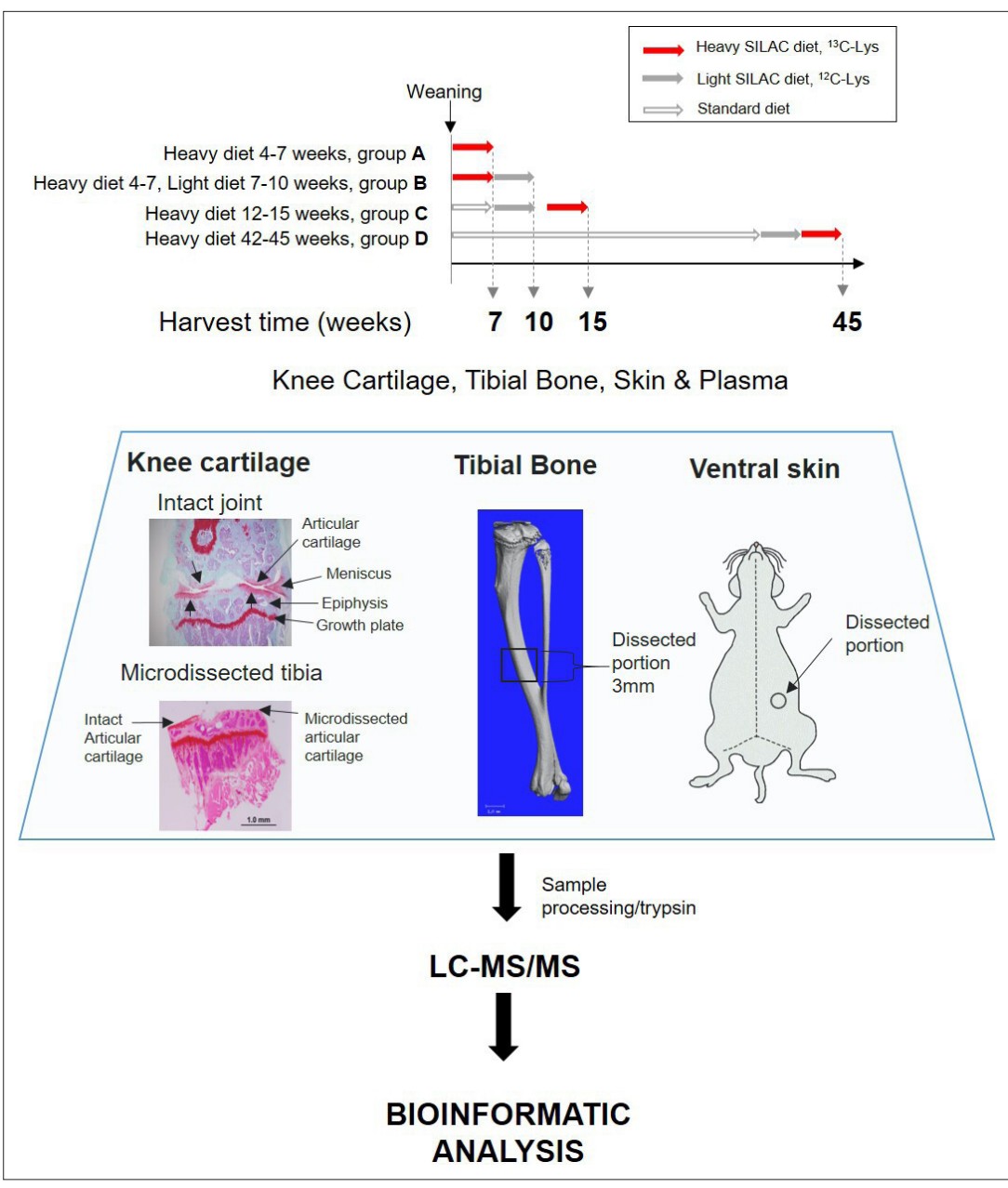

**Figure 1.** Experimental design. Four groups of four C57BL/6 J male mice were fed with heavy SILAC diet ($^{13}C_6$-Lys) or light SILAC diet ($^{12}C_6$-Lys) for 3 weeks at different ages. Two groups of mice (A and B) were fed with the heavy diet from weeks 4 to 7. Group A was culled for tissue collection, and group B was switched to light diet from weeks 7 to 10, then culled for tissue collection. Groups C and D were fed with the heavy SILAC diet for 3 weeks until week 15 and week 45 respectively. Plasma, knee articular cartilage, tibial bone, and ventral skin were collected from fixed anatomical positions shown above. Left-hand panel shows Safranin O stained coronal sections of a murine knee joint before and after micro-dissection of the articular cartilage. Tissues were processed according to tissue specific protocols, trypsinised, and peptides analysed by liquid chromatography–tandem mass spectrometry. Peptides and protein identification and heavy/light (H/L) ratios were obtained by Maxquant software.

## Results

### Proteome turnover in cartilage, skin, bone, and plasma during skeletal growth

To measure new protein synthesis and incorporation during the period of maximal skeletal growth, we pulse labelled two groups of mice (groups A and B, *Figure 1*) by feeding them heavy SILAC diet for

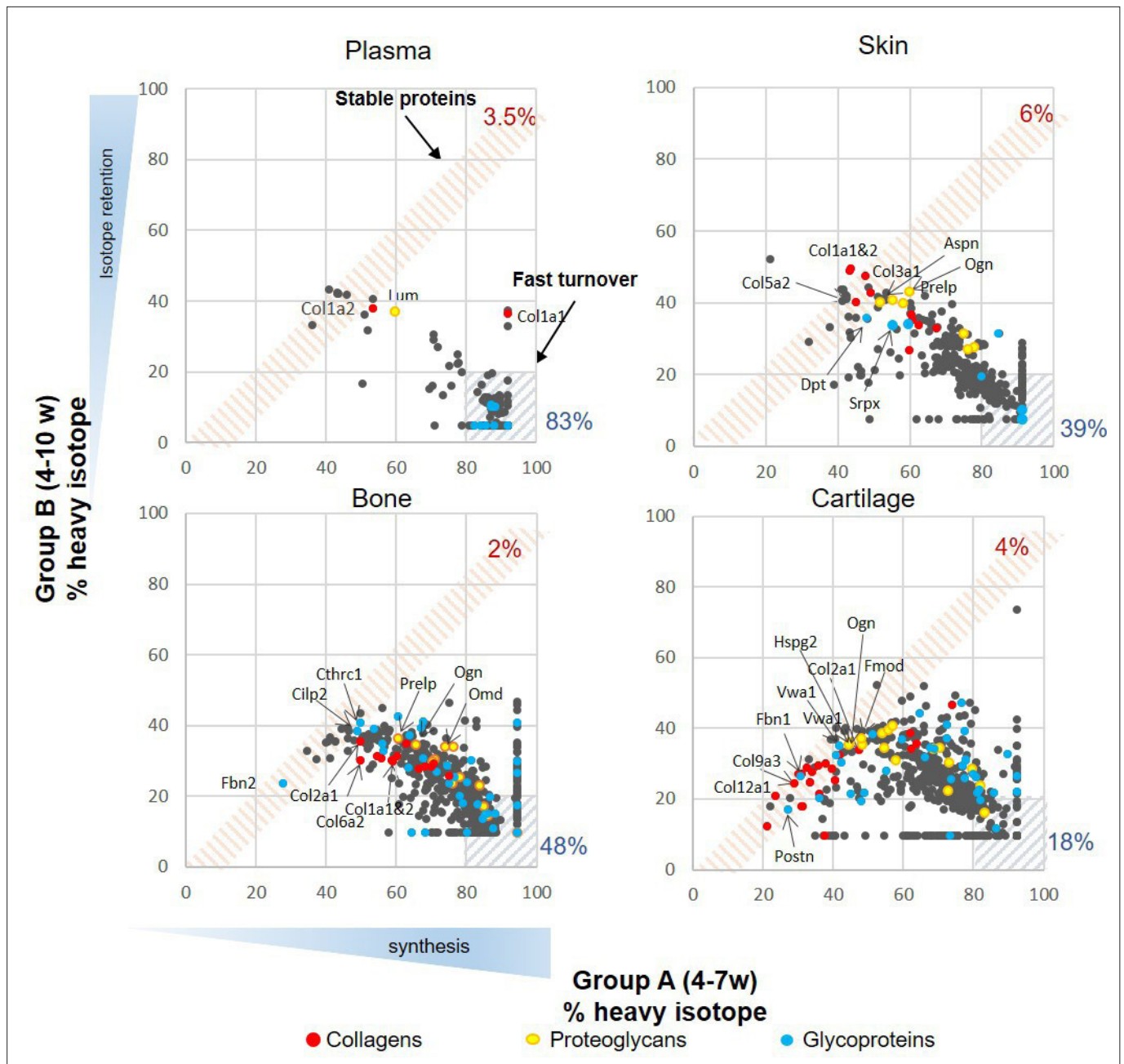

**Figure 2.** Synthesis rate and incorporation of protein in plasma, skin, bone, and articular cartilage during skeletal growth. Each dot represents the mean of the percentage of incorporation of the heavy isotope for an individual protein, n = 4. The x axis represents percentage of the heavy isotope incorporation into proteins from weeks 4 to 7 of age (group A), and the y axis the heavy isotope subsequently lost during the light diet, weeks 7–10 (group B). Collagens are highlighted in red, proteoglycans in yellow, and glycoproteins in blue. The three most stable collagens, proteoglycans, and glycoproteins in each tissue are named. Percentage of stable (red hashed area) and fast turnover proteins (grey hashed areas) for each tissue are indicated.

The online version of this article includes the following figure supplement(s) for figure 2:

**Source data 1.** Full profile of incorporation rates (H/L), iBAQ L and iBAQ H values for plasma, skin, bone, and cartilage protein groups during skeletal growth.

**Figure supplement 1.** Incorporation of heavy isotope in plasma, skin, bone, and cartilage.

3 weeks from 4 weeks of age. Group A was culled for tissue collection immediately after the feeding period, whilst group B was fed the light SILAC diet for another 3 weeks (to week 10). Plasma, knee articular cartilage, ventral skin, and tibial bone were collected for analysis at week 7 (group A) or week 10 (group B). Tissues were extracted and analyzed by mass spectrometry (*Figure 1*).

*Figure 2* shows the percentage of heavy isotope incorporated into individual proteins at the end of the labelling period (week 7) compared with the amount of labelled protein remaining in that tissue after a 3 week washout period of light isotope diet (week 10). In essence, reflecting proteins that are newly incorporated into the tissue and their stability over time. Each tissue was considered separately. We used the plasma profile to define a fast turnover protein group as previously described (*Price et al., 2010*). As the animals are growing during this period, the amount of label remaining after the 3 week washout period reflects both protein turnover and label dilution due to tissue expansion. Proteins that showed over 80 % incorporation during the heavy diet period and retained less than 20 % after the light diet period were considered fast turnover proteins (hashed blue area, *Figure 2*), while proteins that showed little change in the percentage of isotope incorporation between the end of the heavy diet (week 7) and the end of the light diet (week 10) were regarded as 'stable proteins' (hashed pink area, *Figure 2*). Over 83 % (142/171) of plasma proteins fell within the fast turnover group, while only 3.5% (6/171) proteins were within the stable group. Cartilage showed the lowest proportion of fast turnover proteins (18%, 115/634) when compared with bone (48%, 338/712) and skin (39%, 136/352), and was also the tissue that contained the most proteins with less than 40% isotope incorporation during the heavy diet period (bottom left-hand corner of graphs, *Figure 2*). The patterns of heavy isotope incorporation were similar across the four biological replicates for each

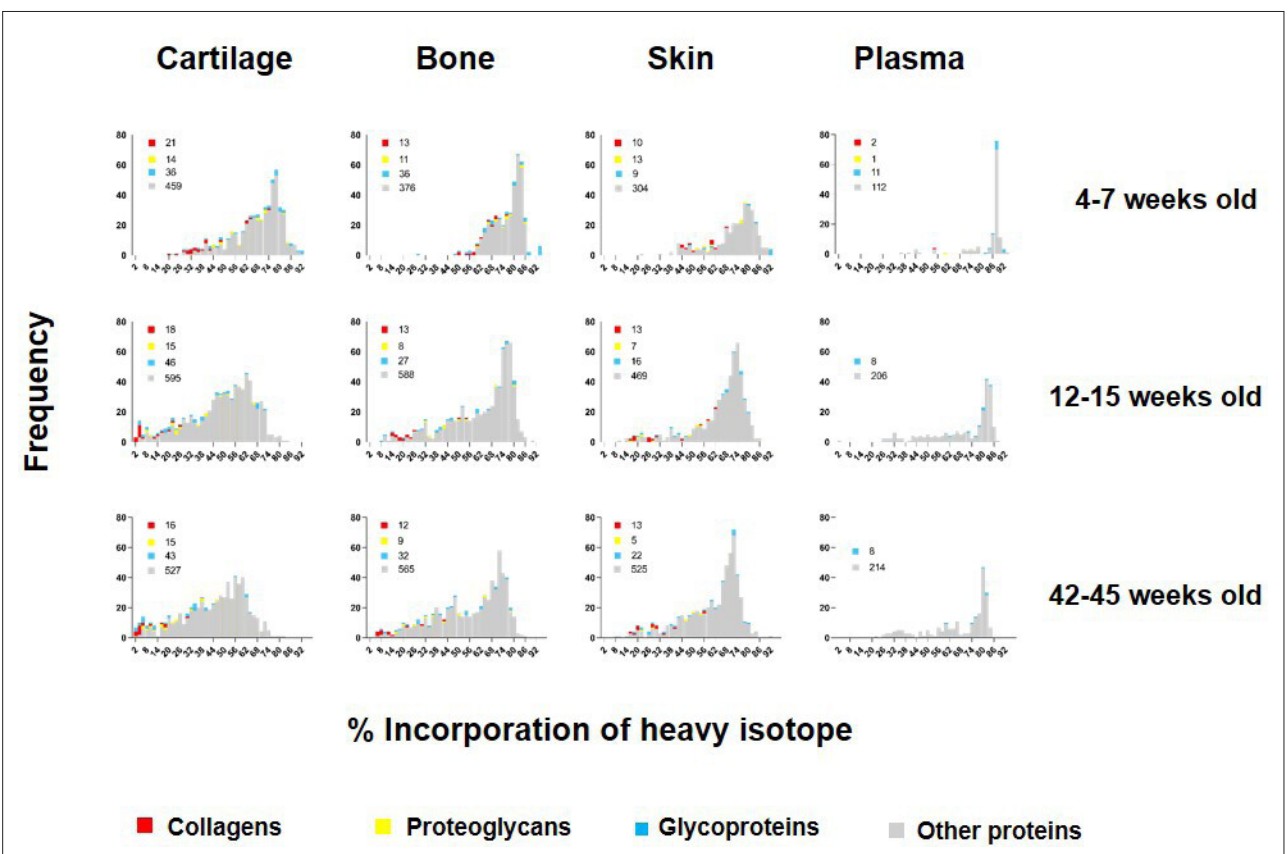

**Figure 3.** Percentage of incorporation of heavy isotope into newly synthesised proteins in plasma, ventral skin, knee articular cartilage, and tibial bone at different ages. The percentage of newly synthesised proteins during each heavy diet period is reflected by the percentage of heavy isotope ($^{13}C_6$-Lys) incorporated into proteins. Collagens are labelled in red, proteoglycans in yellow, glycoproteins in blue, and other proteins in grey. Corresponding numbers of a chains are located next to each colour code in legends. New protein synthesis was significantly different between the four tissues and at all age groups, Kruskal–Wallis test, p<0.0001 for all comparisons.

tissue and time point (*Figure 2—figure supplement 1*). The full profile of the four tissue proteomes during skeletal growth are presented in *Figure 2—source data 1*.

When considering the stability of proteins across the whole proteome by examining the ratio (fold change) of % heavy label for each protein at time B compared with time A, a broad range of protein stabilities was demonstrated in each of the tissues (fold change range 0.05–1.06 [plasma], 0.11–1.0 [cartilage], 0.10–0.95 [bone], and 0.08–1.14 [skin]). Of these, matrisome proteins dominated the most stable proteins (ratios ≥ 0.8) particularly in cartilage and skin (12/18 proteins in skin, 14/25 proteins in cartilage, 2/14 proteins in bone, and 0/6 in plasma). The three most stable collagens (red dots), proteoglycans (yellow dots), and glycoproteins (blue dots) are highlighted for each tissue (*Figure 2*). During this period of skeletal growth, collagens exhibited variable stability across all tissues, and this included the fibrillar collagens (types I, II, III, V, and XI), which are generally regarded as being the most stable. Proteoglycan synthesis was more dynamic, showing lower fold change levels of the B/A ratio. Glycoproteins spanned a wide turnover range in all tissues.

## Age-dependent remodelling of the tissue proteome

To investigate how new protein synthesis and incorporation changes during ageing, we pulse labelled another two groups of four male mice: groups C and D. Group C represented skeletally mature (young adult) mice fed heavy diet between 12 and 15 weeks of age. Group D mice were fed heavy diet between 42 and 45 weeks of age (older adult). For each animal, plasma, knee articular cartilage, ventral skin, and tibial bone were processed immediately after the 3 week heavy diet as in A (*Figure 1*). Histograms showing the frequency of incorporation of the heavy isotope into proteins in groups A, C, and D are shown in *Figure 3*. Protein synthesis generally decreased in all four tissues with age, as shown by a shift to the left of the histograms. As expected, this shift was more apparent for bone and cartilage between skeletally immature (group A) and young adult animals (group C) than between young (group C) and older adult mice (group D), commensurate with cessation of skeletal maturation. Although all tissues showed an age-related decline in new protein incorporation, protein synthesis was significantly different between the four tissues and at each age group, Kruskal–Wallis test, $p < 0.0001$ for all comparisons. As expected, collagens (shown in red, *Figure 3*) dominated the proteins incorporating at the lowest levels (lowest % label).

We were able to determine protein synthesis rates in 27 different collagens α chains (*Figure 4*, *Figure 4—source data 1*), 21 different proteoglycans (*Figure 5*, *Figure 5—source data 1*), and 66 glycoproteins (*Figure 6*, *Figure 6—source data 1*) across all tissues and age groups. Fibrillar collagens (I, II, III, V, and XI) showed the greatest decline of incorporation rates between skeletal immaturity and maturity, compared with non-fibrillar: FACIT (fibrillar associated with interrupted triple helices) (IX, XII, XIV, XVI, XXII), network (IV, X), and beaded filament (VI) collagens. Major collagens, type II in cartilage and type I in bone and skin, had the most striking decrease in new incorporation, falling to <2% in cartilage upon reaching skeletal maturity. Conversely, a small number of matrisome proteins were incorporated for the first time in adult tissues such as versican in cartilage and type XV collagen in bone. Compared with collagens, proteoglycans and glycoproteins maintained a higher incorporation rate across the healthy life course.

To examine changes in new protein incorporation across the whole tissue proteome with age, we compared the whole proteome profile in skeletally mature groups D and C. Data are represented in volcano plots (*Figure 7*). Significantly regulated proteins with –1.5 > FC > 1.5, and with Benjamini–Hochberg (BH) correction FDR p<0.05, for each tissue, are highlighted (bold points). Overall, of those proteins whose incorporation changed with age, most showed reduced incorporation. In connective tissues these were 37 from 452 in skin, 291 from 572 in bone, and 175 from 597 in cartilage. In plasma, 83 of 180 proteins exhibited statistically significant regulation upon ageing (proteins shown in *Figure 7—source data 1*). A small number of proteins had increased incorporation with age. Only two regulated proteins were common among the three collagenous tissues; H1f0, a histone protein, down-regulated in cartilage and bone but upregulated in skin, and Rpl12, a structural constituent of the ribosome 60 S subunit, down-regulated in the three tissues. Articular cartilage and tibial bone proteomes shared 9.6 % of the statistically significant, regulated proteins while overlap between bone and skin was 1.1%, and cartilage and skin 0.7 %. Total protein levels in each tissue, assessed by iBAQ measurements, remained unchanged post-skeletal maturity with just seven proteins changing significantly in articular cartilage (*Figure 7—figure supplement 1*). This likely reflects the substantially enhanced sensitivity of the pulse SILAC methodology to detect alterations in protein homeostasis.

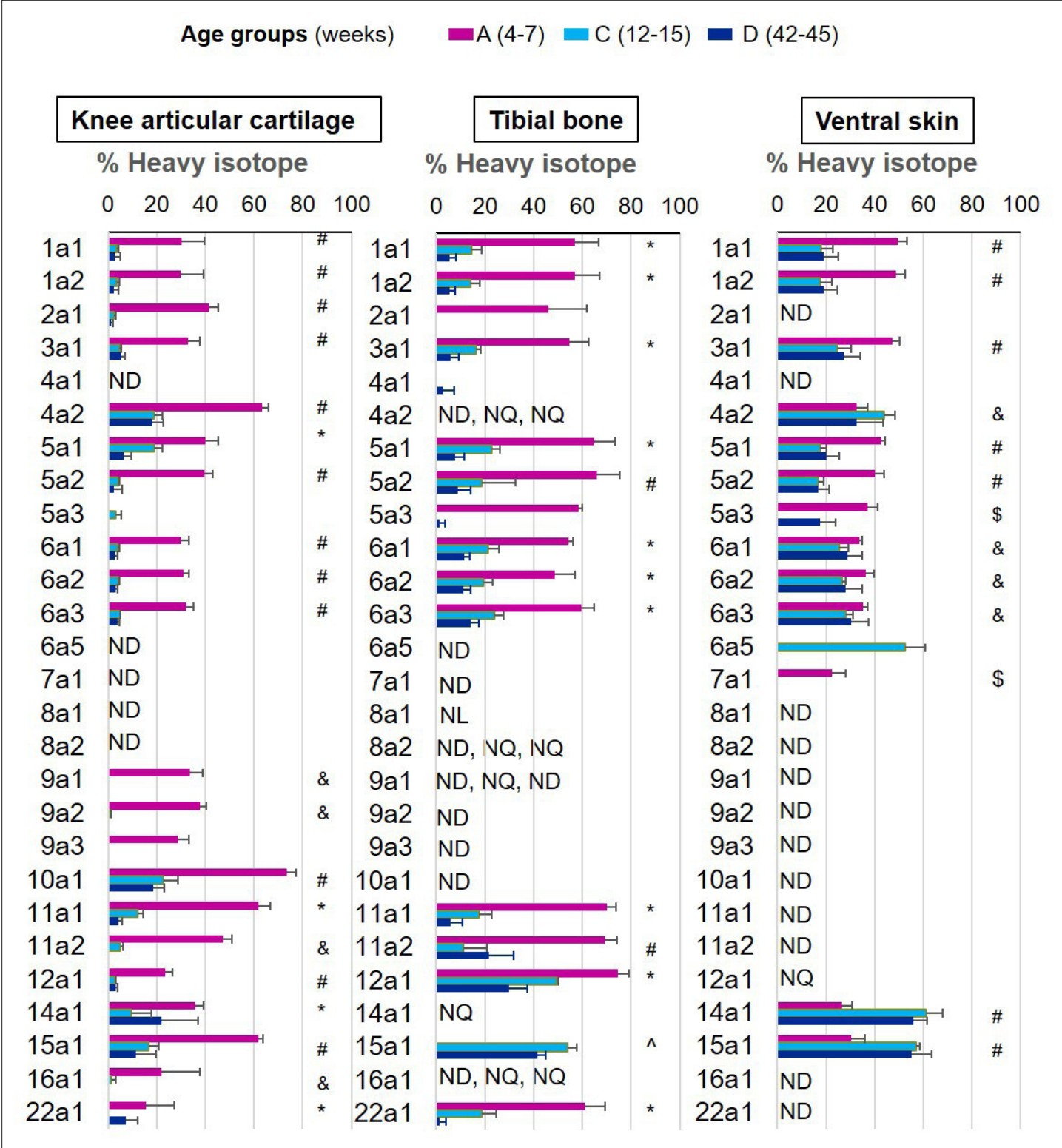

**Figure 4.** New collagen incorporation into different tissues during ageing. The percentage of newly synthesised collagen incorporated into articular cartilage, tibial bone, and ventral skin is estimated by the incorporation of the heavy isotope ($^{13}C_6$-Lys) into proteins during the 3 weeks of heavy diet. The three periods of heavy diet were compared: skeletal growth (4–7 weeks old), young adults just after skeletal maturity (12–15 weeks old), and older adults (42–45 weeks old). Error bars represent the standard deviation (n = 4). ND = not detected, protein group not present in the group dataset. NQ = not quantified, protein group present, but with <2 samples quantified. NL = not labelled, protein group with only light isotope quantified. Statistical significance was determined by pairwise comparisons using Perseus software, Student's t-test reporting Benjamini-Hochberg adjusted p<0.05.

*Figure 4 continued on next page*

*Figure 4 continued*

Statistically significant differences were denoted as follows: &between groups A and C, $between groups A and D, ^between groups C and D, #between groups A–C and A–D, *between all three groups.

The online version of this article includes the following figure supplement(s) for figure 4:

**Source data 1.** Data supporting *Figure 5*.

STRING protein interaction networks of statistically significant, regulated proteins for the four tissues are shown in *Figure 7*, with full list of cluster elements in *Figure 7—source data 2*. Each of the tissues shared a common cluster relating to regulatory elements involving ribosomal and histone proteins, albeit with different constituent proteins in each tissue. The largest cluster of interaction networks in cartilage (*Figure 7*) was mainly formed by ECM proteins such as collagens, collagen processing-related, and proteoglycans. Also notable, were two clusters abundant in proteoglycans and growth factors, and another cluster rich in myosin family members. Fifteen proteins with decreased incorporation have previously been implicated in OA development (*Figure 7—source data 2*, final column).

In bone, the largest cluster represented several myosin family members, involved in muscle contraction, some structural proteins, and proteins with roles in ATP transfer. Two unique bone clusters were involved in energy metabolism; one formed by 29 mitochondrial proteins such as ATP synthases, cytochrome oxidases, and NADH dehydrogenases, and the other largely by enzymes involved in glycolysis and fatty acid metabolism (*Figure 7*, *Figure 7—source data 2*). Five proteins with a statistically significant decrease in incorporation have previously been linked to osteoporosis including biglycan, Col1a2 and troponin C2, YWHAE and Cd44 (*Figure 7—source data 2*, final column).

One striking result was the relative paucity of the percentage of age-regulated proteins in skin (8%) compared with cartilage (29%) and bone (51%) despite a similar number of total labelled proteins quantified (*Figure 7—figure supplement 2*). In skin, a number of weak clusters were identified with a couple involving energy metabolism (*Figure 7*, *Figure 7—source data 2*). In plasma, one strong cluster comprising 40 proteins with roles in lipid transport (apolipoproteins), proteins involved in the innate immune response, and several serpin family members, was identified (*Figure 7*, *Figure 7—source data 2*). Further bioinformatic pathway analysis was performed using Database for Annotation, Visualization and Integrated Discovery (DAVID), which produced similar results to STRING (data not shown), and Ingenuity Pathway Analysis (IPA). IPA identified a number of enriched cellular pathways (*Figure 7—figure supplement 2*), but none of these reached corrected statistical significance.

## Discussion

To extend our knowledge of connective tissue renewal during growth and ageing, we used metabolic labelling with stable isotopes in combination with quantitative proteomics to estimate protein incorporation rates in live animals across the healthy life course. Here, we describe tissue proteome incorporation rates for proteins in skin, cartilage, bone, and plasma at three post-natal stages of life: during maximum skeletal growth, and at young and older adulthood. Our study shows that although new protein incorporation changes significantly in all tissues after skeletal maturity, it displays distinct temporal and molecular tissue signatures.

Tissue turnover can be divided into tissue modelling, which occurs during growth and development, and tissue remodelling, which is required to repair or adapt the tissue to maintain homeostasis (*Frantz et al., 2010*; *Karsdal et al., 2016*). Collagen-rich tissues are believed to have a limited reparative capacity, mainly due to the inability to incorporate new fibrillar collagens into mature matrices (*Verzijl et al., 2000*). Our results quantified incorporation of multiple collagens in each of the connective tissues, with almost all of them showing decreases in incorporation with age. This was most marked for bone and cartilage. These tissues are largely post-mitotic after skeletal maturity, and although they retain global synthetic activity in adult life, this is blunted in a sizeable fraction of proteins (29 % bone, 51 % cartilage) by 45 weeks of age compared with skin (8%), in which the tissue is known to renew throughout adult life (*Visscher et al., 2015*). Bone was the only tissue that showed a strong collagen cluster (enriched in 12 collagens) when comparing young with older adult tissue, although, as expected, incorporation of new type II collagen fell to <2% in cartilage in the older group. In skin, significant decreases were also observed in fibrillar collagens, but the transition between skeletally

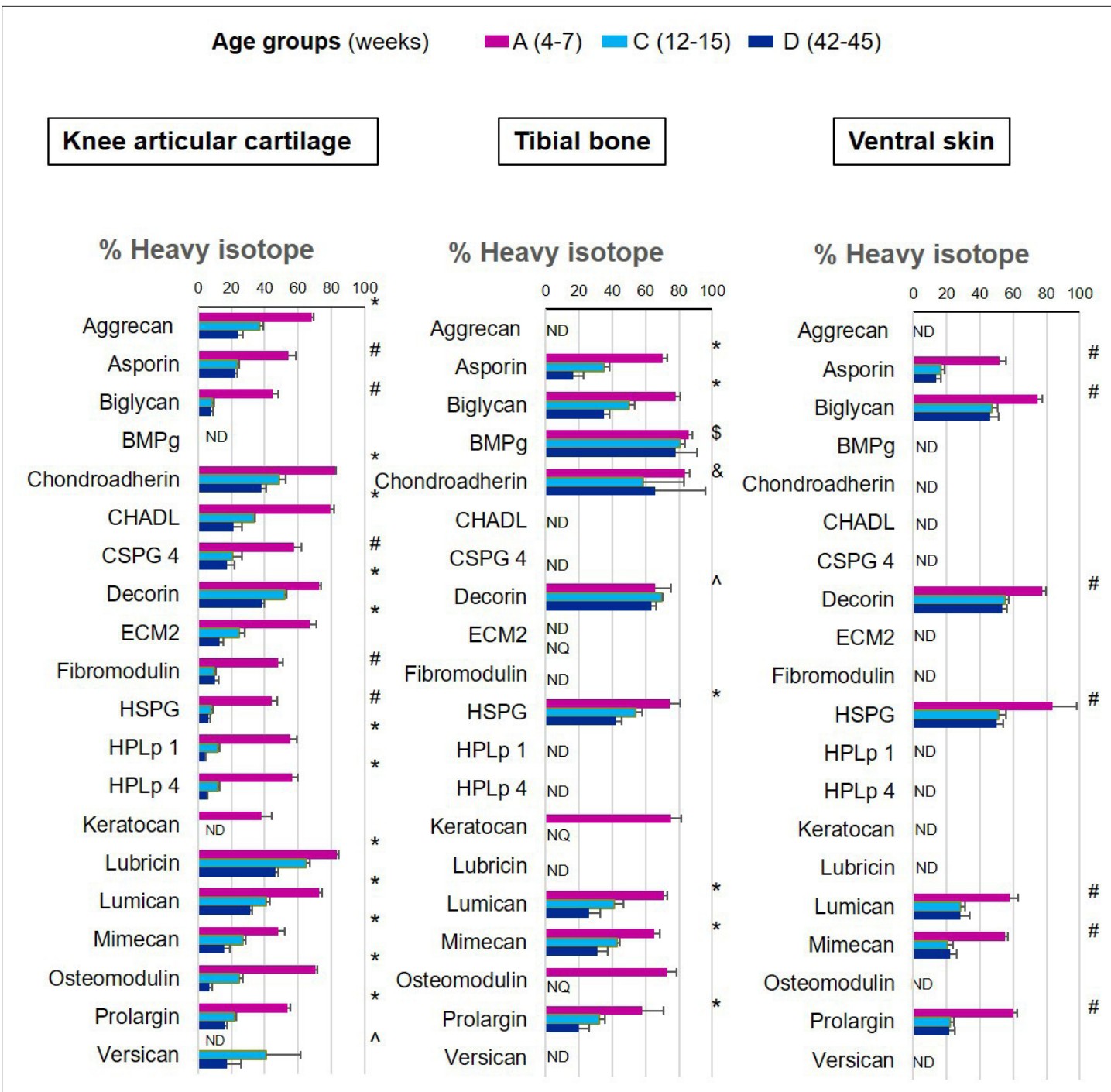

**Figure 5.** New proteoglycan incorporation rate into different tissues during ageing. The percentage of newly synthesised proteoglycans incorporated into articular cartilage, tibial bone, and ventral skin is estimated by the percentage of incorporation of the heavy isotope ($^{13}C_6$-Lys) into the proteins during the 3 weeks of heavy diet. Protein synthesis and incorporation was estimated across the healthy life span, covering skeletal growth (4–7 weeks old), young adults just after skeletal maturity (12–15 weeks old), and older adults (42–45 weeks old). Error bars represent the standard deviation (n = 4). ND = not detected, protein group not present in the group dataset. NQ = not quantified, protein group present, but with <2 quantified samples. BMPg, bone marrow proteoglycan. Statistical significance was determined by pairwise comparisons using Perseus software, Student's t-test reporting Benjamini-Hochberg adjusted p<0.05. Statistically significant differences were denoted as follows: &between groups A and C, $between groups A and D, ^between groups C and D, #between groups A–C and A–D, *between all three groups.

The online version of this article includes the following figure supplement(s) for figure 5:

**Source data 1.** Data supporting *Figure 5*.

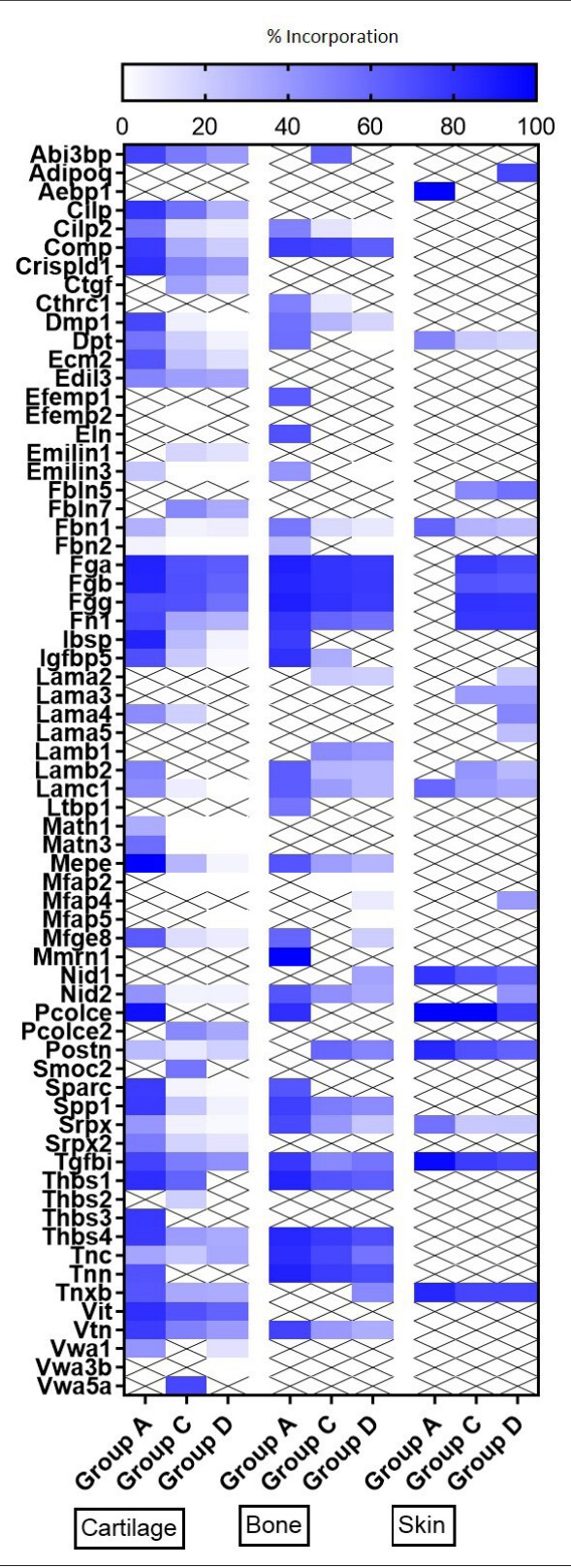

**Figure 6.** Heatmap of new glycoproteins incorporation rates into different tissues during ageing. The percentage of newly synthesised glycoproteins incorporated into articular cartilage, tibial bone, and ventral skin is estimated by the percentage of incorporation of the heavy isotope ($^{13}C_6$-Lys) into the proteins during the 3 weeks of heavy diet. Protein synthesis and incorporation was estimated across life, covering skeletal growth (4–7 weeks old), young

*Figure 6 continued on next page*

*Figure 6 continued*

adults just after skeletal maturity (12–15 weeks old), and older adults (42–45 weeks old).

The online version of this article includes the following figure supplement(s) for figure 6:

**Source data 1.** Data supporting *Figure 6*.

immature and mature tissues was less marked, with substantial on-going incorporation of fibrillar collagens through young and older adulthood.

Previous age-related research has identified several cellular pathways that are consistently found to be dysregulated. Cell energy and metabolism, including mitochondrial dysfunction, is one of the top processes (*Campisi et al., 2019*; *Javadov et al., 2020*; *Sprenger and Langer, 2019*). Proteins such as clusterin, superoxide dismutase 1 (SOD1), and apolipoprotein E have previously been found to be dysregulated in ageing (*Sabaretnam et al., 2010*; *Sentman et al., 2006*; *Trougakos and Gonos, 2006*). Although we were underpowered to gain significant insights from IPA and DAVID pathway analysis packages, by STRING analysis bone displayed a very strong metabolic, mitochondrial signature, and we found clusterin dysregulation in plasma, cartilage, and bone of older adult mice, and down-regulation of SOD1 in cartilage. Modulators of protein synthesis such as heterogeneous nuclear ribonucleoprotein D, which affects stability and metabolism of mRNAs, were also down-regulated in cartilage and have previously been linked to ageing (*Pont et al., 2012*). Our data showed only two proteins where reduced incorporation was seen in all three collagenous tissues. These included the ribosomal protein, RPL12 and Histone H1.0. Moreover, strong down-regulated clusters, rich in ribosomal proteins, were present in bone, cartilage, and skin, indicating a decline in incorporation rates of ribosomal structural proteins across all tissues as the animal ages. This finding agrees with several studies in which progressive decreases in the expression of ribosomal proteins or rRNA has been observed with age (*D'Aquila et al., 2017*; *Jung et al., 2015*).

In cartilage, 15 of the down-regulated proteins have been associated with OA. Several of these have known protective functions in cartilage tissue homeostasis, such as Prg4 (also known as lubricin) (*Flannery et al., 2009*), Timp3, an inhibitor of disease-associated metalloproteinases (*Nakamura et al., 2020*), and LDL receptor-related protein 1 (Lrp1), an important chondrocyte scavenger receptor (*Yamamoto et al., 2013*). In addition, a number of pro-repair and chondrogenic growth factors, all known to be sequestered in the pericellular matrix, were reduced in older adult cartilage such as fibroblast growth factor 2 (FGF2), connective tissue growth factor, transforming growth factor beta (TGFβ1), and Frzb, a Wnt antagonist (*Chia et al., 2009*; *Tang et al., 2018*; *Khan et al., 2011*; *Van Der Kraan, 2017*; *Evangelou et al., 2009*; *Huang et al., 2019*; *Zhong et al., 2017*). These results point towards loss of repair pathways in the pathogenesis of age-related OA, a conclusion supported by recent genome-wide association studies (*Tachmazidou et al., 2019*).

The balance between bone formation and resorption is compromised with increasing age, resulting in progressive net bone loss and osteoporosis (*Javaheri and Pitsillides, 2019*). Three proteins whose incorporation rates decreased with age are regarded as biomarkers of osteoporosis. One of these was biglycan, a bone modifying protein in murine osteoporosis (*Cui et al., 2019*) and collagen 1a2, the most abundant protein in bone matrix (*Nagy et al., 2020*). In addition, we found decreased incorporation in two proteins, 14-3-3 protein epsilon and Cd44. SNPs in these two protein coding genes, *YWHAE* and *CD44*, are associated with low bone mineral density in two recent GWAS studies (*Morris et al., 2019*; *Medina-Gomez et al., 2018*).

In skin, only 8 % of proteins had altered incorporation rates in the older group, considerably lower than in cartilage and bone (29% and 51%, respectively). Six clusters comprising a maximum of four proteins in each were discerned. Wound healing is known to be impaired in the elderly adults, but this phenotype is probably less apparent in 'middle aged' individuals which would better represent our older adult mice (*Gardiner, 2011*). One down-regulated protein in the skin, solute carrier family 3 (SLC3A2) (*Figure 7—source data 1*), has been implicated in wound healing (*Falanga, 2005*; *Grande-Garcia et al., 2007*; *Boulter et al., 2013*), and ageing (*Tissot et al., 2018*). Although there were several proteins with statistically significant differences in incorporation rates in older skin, the clusters did not predict an overall negative effect on wound healing although this tissue signature might not be expected in non-wounded tissues.

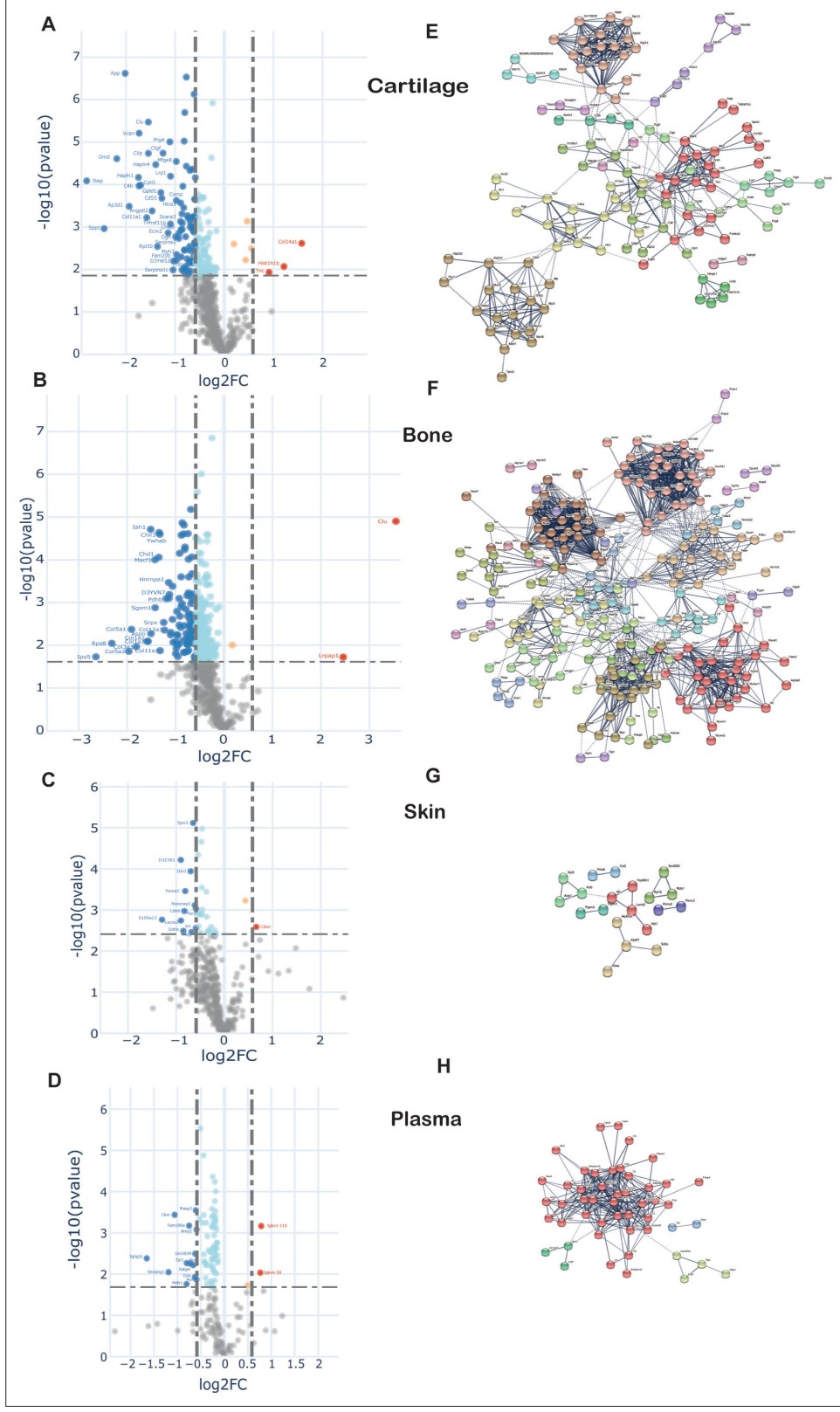

**Figure 7.** Changes in protein synthesis and incorporation rates during tissue remodelling. (**A–D**) Differential protein incorporation rates in young (15 weeks) versus older (45 weeks) adult tissues. (**A**) Cartilage, (**B**) bone, (**C**) skin, and (**D**) plasma. Volcano plots, unpaired Student's t-test with BH correction FDR < 0.05, FC > 1.5, n = 4. Full list of proteins available in *Figure 7—source data 1*. (**E–H**) STRING cluster analysis of differentially incorporated

*Figure 7 continued on next page*

*Figure 7 continued*

proteins in each tissue. (**E**) Cartilage, (**F**) bone, (**G**) skin, and (**F**) plasma. Edges show high confidence interaction score = 0.7. Networks clustered to MCL inflation parameter = 2. Cluster elements are listed in *Figure 7—source data 2*.

The online version of this article includes the following figure supplement(s) for figure 7:

**Source data 1.** Data supporting *Figure 7A–D*.

**Source data 2.** Data supporting *Figure 7E–H*.

**Figure supplement 1.** Changes in protein abundance from young to older adults.

**Figure supplement 2.** Protein overlap between the four tissues.

**Figure supplement 3.** Pathway enrichment using by IPA for young versus older adult regulated protein profiles.

We recognise a number of limitations in this study. For cartilage and bone, an insoluble pellet remained after sample processing despite multiple refinements to the protocol for each tissue. It is likely that the insoluble fraction is mainly composed of the oldest, highly cross-linked portion of fibrillar collagens that increases with age (*Robins, 2007*; *Verzijl et al., 2000*). Therefore, in the older group, the proportion of newly synthesised collagens might be overestimated. It is also the case that the insoluble pellet may have been influenced by a change in the ratio of cortical to trabecular bone with age. Another limitation was that we only studied mice until 45 weeks of age, approximately half the lifetime of a laboratory mouse. We did not include older mice because they frequently develop spontaneous OA, which would have been a co-founding variable in cartilage, preventing us from discerning effects due to age and those due to disease. Likewise, in bone, insidious loss of tissue density occurs with age (*Jilka, 2013*). In this study, we only examined male mice to keep our initial analysis simple. There is a clear gender difference in the risk of OA and osteoporosis in both humans and mice, so a male/female comparison would be a valuable future proposal to uncover important sex-dependent biological differences that contribute to disease risk. Finally, we limited this analysis to only three contrasting connective tissues, each of which is distinct in terms of cell type and number. As the labelled mouse carcasses are currently frozen, the authors welcome collaborative approaches to extend this analysis to other tissues in line with the three R principles (*Percie du Sert et al., 2020*).

Compared with protein abundance, which only captures a snapshot of a protein at a particular time, this methodology provides insights into dynamic protein synthesis in different tissues and at different ages. Whilst we do not measure the stability of individual proteins, we reveal distinct age-dependent synthetic activities in dividing and post-mitotic connective tissues and identify shared and tissue-specific protein clusters that could represent novel insights into age-dependent disease.

## Materials and methods

**Key resources table**

| Reagent type (species) or resource | Designation | Source or reference | Identifiers | Additional information |
|---|---|---|---|---|
| C57BL/6 J (*Mus musculus*, male) | C57BL/6 J | 4 weeks old mice were from in house breeding colony and 11 and 41 weeks old from Charles rivers laboratories, Oxford, England | RRID:MGI:5655520 | Wild type |
| Chemical compound, drug | $^{13}C_6$-Lysine-SILAC (97 % atom $^{13}C$) | SILANTES GmbH, Munich, Germany | 13C-aa-Lys(6)-SILAC-Mouse diet-252923926 | |
| Chemical compound, drug | $^{12}C_6$-Lysine-SILAC diets | SILANTES GmbH, Munich, Germany | 12C-aa-Lys(0)-SILAC-Mouse diet-230004600 | |
| Software, algorithm | GraphPad Prism 7 | GraphPad Software Inc, San Diego, CA, USA | Prism: RRID:SCR_015807 | GraphPad Prism 7 |
| Software, algorithm | MaxQuant | MaxQuant | MaxQuant: RRID:SCR_014485 | MaxQuant v1.5.7.4 |
| Software, algorithm | Perseus | Perseus | Perseus: SCR_015753 | Perseus v1.6.1.1 |

*Continued on next page*

*Continued*

| Reagent type (species) or resource | Designation | Source or reference | Identifiers | Additional information |
| --- | --- | --- | --- | --- |
| Software, algorithm | Clinical Knowledge Graph's analytics core | CKG | | |
| Software, algorithm | STRING | STRING | STRING: SCR_005223 | STRING v11 |
| Software, algorithm | IPA | QIAGEN Inc | QIAGEN IPA: SCR_008653 | |
| Software, algorithm | The Database for Annotation, Visualization and Integrated Discovery (DAVID ) v6.8 | DAVID | | DAVID v6.8 |

## Samples and reagents

All animal experiments were carried out with Institutional Ethical Approval Under Animals (Scientific Procedures) Act 1986 (Licence Number 30-3129) Wild-type C57BL/6 J mice, ages 4, 11, and 41 weeks, were obtained originally from Charles River, Oxford, England. Mice from experimental groups A and B were taken from a breeding colony of these mice within our animal unit. Groups C and D were purchased directly for the experiment. $^{13}C_6$-Lysine-SILAC (97 % atom $^{13}C$) (heavy diet) and $^{12}C_6$-Lysine-SILAC (light diet) were purchased from SILANTES GmbH, Munich, Germany. Animals' care was in accordance with institutional guidelines.

### Pulsed SILAC labelling experimental design

Our labelling strategy was aimed at labelling long lived ECM proteins. Therefore, four groups of four male mice were fed with a stable isotope diet for a period of 3 weeks at different ages that spanned from weaning to late adulthood. The labelling scheme (*Figure 1A*) was as follows: Two groups of mice (A and B) were fed with the heavy diet from weeks 4 to 7. Then, one of these groups was culled for tissue collection and the other was changed to the light isotope diet ($^{12}C_6$-Lys) from weeks 7 to 10, then culled for tissue collection. This was used to calculate proteome turnover rates during skeletal growth. The third group (C) was fed with the heavy isotope diet from weeks 12 to 15 (young adult), and the fourth group (D) was fed with the heavy isotope diet from weeks 42 to 45 (late adult). All mice were acclimatised to the SILAC diet formulation by feeding them with the light isotope diet for 1 week before introducing the heavy diet. All mice gained weight in accordance with reference laboratory data for C57BL/6 mice (data not shown).

### Tissue harvest

At the end of each labelling period, corresponding to ages 7, 10, 15, and 45 weeks, animals were culled injecting a terminal dose (30 mg/animal) of the anaesthetic Pentoject. Blood, knee articular cartilage, tibial bone, and ventral skin were collected from each animal (n = 4 mice per time point) as follows: blood was collected by cardiac puncture and mixed with 0.5 M anticoagulant EDTA to obtain a final concentration of 5 mM EDTA. The buffered blood was centrifuged for 15 min at 3000 rpm at 4 °C. The resulting supernatant (plasma) was transferred to a 0.5 ml tube to be used for analysis. Knee articular cartilage was harvested using a micro-dissection technique previously developed described by our group (*Gardiner, 2011*). Articular cartilage from the femoral and tibial surfaces of both knees of one mouse were micro-dissected under the stereo microscope and collected in 1.5 ml micro-centrifuge tubes containing 50 µl PBS. Tibial bone samples were sectioned between the crest of the tibia and the insertion of the fibula. The bone marrow was flushed three times with PBS to remove cells using a 25 G needle. The final wash was checked for lack of cells using a stereo microscope. Approximately 0.8 cm$^2$ of skin was cut from the lower flank of the ventral surface. Hair was removed using hair removal cream (Veet sensitive skin). Adipose tissue and blood vessels were removed from the subcutaneous region under the stereo microscope, and a portion of clean skin was collected using a 6 mm biopsy punch. All tissues were washed with phosphate-buffered saline (PBS [pH 7.4]) and stored frozen at −80 °C until further processing.

## Mass spectrometry sample preparation

Bone and cartilage samples were placed in 180 µl of 5 mM dithiothreitol (DTT) and heated at 65 °C for 15 min. After cooling to room temperature, samples were alkylated with 20 mM iodoacetamide (IAA) for 30 min. To quench remaining IAA, DTT was added and samples were incubated for 30 min. Then, samples were incubated with 4 M GuHCl for 2 hr. The samples were adjusted to pH 8 with 400 mM Tris base. Proteins were digested with 1 µg of trypsin overnight at 37 °C. Digestion was terminated adding trifluoroacetic acid (TFA) to a final concentration of 0.5–1%. Peptides were purified using C18 solid phase extraction cartridges (SOLA HRP SPE cartridges, Thermo Fisher Scientific).

Five microlitres of plasma was reduced with 5 mM DTT, alkylated with 20 mM IAA, and proteins precipitated with methanol/chloroform (*Wessel and Flugge, 1984*). Precipitated proteins were solubilised in 6 M urea buffer. Urea was diluted to <1 M with milli-Q water and proteins digested with trypsin overnight at 37 °C at an enzyme to substrate ratio of 1:25. Digestion was terminated adding TFA to a final concentration of 0.5–1%. Peptides were purified using C18 solid-phase extraction cartridges as above.

Skin samples were grinded in a Cryomill, then incubated in 8 M urea, 3 % SDS with protease inhibitors (cOmplete Mini EDTA-free Protease Inhibitor Cocktail, Roche) at room temperature for 1 hr in a total volume of 3 ml. As there was still a substantial pellet, samples were shaken overnight at 4 °C to solubilise the pellet. Samples were centrifuged at 2500 *g* for 10 min and 100 µl of sample was taken for further processing. Proteins were reduced with 5 mM DTT for 30 min at room temperature, alkylated with 20 mM IAA for 30 min at room temperature, and precipitated with methanol/chloroform. Precipitated proteins were solubilised in 6 M urea buffer. Urea was diluted to <1 M with milli-Q water and proteins digested with trypsin overnight at 37 °C at an enzyme to substrate ratio of 1:25. Digestion was terminated adding TFA to a final concentration of 0.5–1%. Peptides were purified using C18 solid-phase extraction cartridges as above.

## Liquid chromatography–tandem mass spectrometry (LC–MS/MS)

Tissues from groups A and B as well as skin samples were analysed on a LC–MS/MS platform consisting of Orbitrap Fusion Lumos coupled to an UPLC ultimate 3000 RSLCnano (Thermo Fisher Scientific) and samples from groups C and D with similar platform but coupled to a Q-Exactive HF. Samples were loaded in 1 % acetonitrile and 0.1 % TFA and eluted with a gradient from 2% to 35 % acetonitrile in 0.1 % formic acid and 5 % DMSO in 60 min with a flow rate of 250 nl/min on a 50 cm EASY-Spray column (ES803, Thermo Fisher Scientific).

## Orbitrap Fusion Lumos

The survey scan was acquired at a resolution of 120,000 between 400 and 1500 m/z and an AGC target of 4E5. Selected precursor ions were isolated in the quadrupole with a mass isolation window of 1.6 Th and analysed after CID fragmentation at 35 % normalised collision energy in the linear ion trap in rapid scan mode. The duty cycle was fixed at 3 s with a maximum injection time of 35 ms, AGC target of 4000, and parallelisation enabled. Selected precursor masses were excluded for the following 27 s.

## Q-Exactive HF

The survey scan was acquired at a resolution of 60,000 between 375 and 1500 m/z with an AGC target of 3E6, up to the top 12 most abundant ions were selected for fragmentation from each scan. Selected precursor ions were isolated with a mass isolation window of 1.2 Th and fragmented by HCD at 28 % normalised collision energy. Fragment scans were acquired at 30,000 resolution with an AGC target of 5E4 and a maximum injection time of 100 ms. Selected precursor masses were excluded for the following 27 s.

## Protein identification

Raw mass spectral data files were searched using MaxQuant software (V1.5.7.4, *Tyanova et al., 2016b*) using SILAC (Lys6) quantitation. Fixed modification was carbamidomethylation of cysteine, and variable modifications were oxidised methionine, deamidation of asparagine and glutamine, acetylation at protein N-terminal, and hydroxylation of proline. The data was searched against the mouse canonical Uniprot database (29/07/2015). FDR on peptide and protein level were set to 1 %.

Second peptide and 'match between runs' options were enabled, all other parameters were left at default settings. The mass spectrometry proteomics data have been deposited to the ProteomeXchange Consortium via the PRIDE (*Perez-Riverol et al., 2019*) partner repository with the dataset identifier PXD023180. Reviewer account details are as follows: Username: reviewer_pxd023180@ebi.ac.uk, Password: iSH0ppKX.

## Statistical analysis

After MaxQuant analysis, Excel version 1.5. 21.1, GraphPad Prism 7 (GraphPad Software Inc, San Diego, CA), STRING, Perseus software version 1.6.1.1 (*Tyanova et al., 2016a*), DAVID, IPA (QIAGEN Inc), and Python libraries from the Clinical Knowledge Graph's analytics core (*Santos et al., 2020*) were used for data visualisation, statistical analysis, and pathway enrichment and protein network analysis.

Proteins were examined for normality against the Shapiro–Wilk normality test and for variance against the Levene test and demonstrated that >80% of proteins were normally distributed. Pairwise comparisons were therefore by Student's t-tests. For group wise comparisons, where the data were not normally distributed, non-parametric tests were used. For those proteins with a heavy to light ratio (H/L), the percentage of label incorporation was estimated as follows: (H/L/((H/L) + 1)) * 100. To discern protein turnover profiles in tissues, we plotted the percentage of label present in group A versus group B. In those cases where H/L ratios were not reported, we used iBAQ light (iBAQ L) and heavy (iBAQ H) data to impute H/L ratios. Missing iBAQ light in >2 samples in group A was assumed to be a consequence of fully labelled protein while missing iBAQ heavy in >2 samples in group B as a result of complete turnover. H/L ratios from samples with missing iBAQ L values were imputed with the maximum H/L value +0.1 in that column and those with missing iBAQ H values with the minimum H/L value – 0.01 (only proteins with lysine containing peptides were imputed). To define what proteins were differentially incorporated when comparing groups D and C (unpaired t-test), we selected proteins with at least two valid values for H/L, and imputed missing values using the K-nearest neighbour algorithm. Furthermore, to account for full labelling or absence of new protein incorporation, we identified proteins where H/L ratios were reported in one group, but not in any of the four replicates in the other group. In those cases, we used heavy or light iBAQ values present in at least two replicates and imputed H/L ratio using uniform random values near the maximum (maximum, maximum+ std) or the minimum (0, minimum) of the dataset, respectively.

The recovered proteins that showed changes in incorporation between groups were added to the STRING networks, DAVID, and IPA analyses.

The STRING networks were built using high confidence interaction score = 0.7, and networks clustered to MCL inflation parameter = 2 (*Szklarczyk et al., 2019*).

## Acknowledgements

We thank H Liao and N Ternette for useful discussions and advice with proteomics data analysis. We also thank G Wilson for helping with figure and table preparations and revisions. YA-M had a fellowship from the Daphne Jackson trust that was co-funded by the Kennedy Trust for Rheumatology Research and University of Oxford. The study was also supported by the Centre for Osteoarthritis Pathogenesis Versus Arthritis (grant nos. 20205 and 21621).

## Additional information

### Funding

| Funder | Grant reference number | Author |
| --- | --- | --- |
| Daphne Jackson Trust | | Yoanna Ariosa-Morejon |
| Versus Arthritis | Centre for Osteoarthritis Pathogenesis 20205 | Tonia L Vincent |
| Versus Arthritis | Centre for Osteoarthritis Pathogenesis 21621 | Tonia L Vincent |

| Funder | Grant reference number | Author |
|--------|------------------------|--------|

The funders had no role in study design, data collection and interpretation, or the decision to submit the work for publication.

## Author contributions

Yoanna Ariosa-Morejon, Conceptualization, Formal analysis, Funding acquisition, Investigation, Methodology, Project administration, Visualization, Writing - original draft; Alberto Santos, Data curation, Formal analysis, Software, Validation, Visualization, Writing – review and editing; Roman Fischer, Conceptualization, Formal analysis, Resources, Supervision, Visualization, Writing – review and editing; Simon Davis, Formal analysis, Methodology, Writing – review and editing; Philip Charles, Formal analysis, Visualization, Writing – review and editing; Rajesh Thakker, Investigation, Writing – review and editing; Angus KT Wann, Visualization, Writing – review and editing; Tonia L Vincent, Conceptualization, Funding acquisition, Project administration, Resources, Supervision, Writing – review and editing

## Author ORCIDs

Roman Fischer http://orcid.org/0000-0002-9715-5951
Philip Charles http://orcid.org/0000-0001-5278-5354
Rajesh Thakker http://orcid.org/0000-0002-1438-3220
Tonia L Vincent http://orcid.org/0000-0002-3412-5712

## Ethics

Animals' care and experimentation was performed in strict accordance with the Animal (Scientific Procedures) Act 1986 and institutional guidelines of the University of Oxford. (licence number 30-3129).

## Decision letter and Author response

Decision letter https://doi.org/10.7554/eLife.66635.sa1
Author response https://doi.org/10.7554/eLife.66635.sa2

# Additional files

## Supplementary files

• Transparent reporting form

## Data availability

Proteomics raw data and Maxquant output files have been deposited to the ProteomeXchange Consortium via the PRIDE (Perez-Riverol et al., 2019) partner repository with the dataset identifier PXD023180. The rest of the data generated or analysed are included in the manuscript and supporting files. Source data files have been provided for all figures.

The following dataset was generated:

| Author(s) | Year | Dataset title | Dataset URL | Database and Identifier |
|-----------|------|---------------|-------------|-------------------------|
| Roman F | 2021 | Age-dependent changes in protein incorporation into collagen-rich tissues of mice by in vivo pulsed SILAC labelling | https://www.ebi.ac.uk/pride/archive/projects/PXD023180 | PRIDE, PXD023180 |

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
