## [Decision Letter]

**Acceptance summary:**

Capturing the rate and degree of protein half-life in tissues rich in collagen, proteoglycans and glycoproteins over the life span provides valuable information about how these tissues age and can increase our understanding of age-related disease. Using a Stable Isotope Labeling (SILAC) method to examine new protein Lys6 incorporation rates in three high collage content tissues, the authors show that protein-change is low in older mice in these tissues, but the depth of the data generated provide a detailed examination of the impact of age at a level we have not previously had. This paper would be of interest to a broad range of scientists studying connective tissues in the context of development and ageing.

**Decision letter after peer review:**

Thank you for submitting your article "Age-dependent changes in protein incorporation into collagen-rich tissues of mice by in vivo pulsed SILAC labelling" for consideration by *eLife*. Your article has been reviewed by 3 peer reviewers, one of whom is a member of our Board of Reviewing Editors, and the evaluation has been overseen by Matt Kaeberlein as the Senior Editor. The reviewers have opted to remain anonymous.

Essential revisions:

1. A major weakness is the tissue/cell turnover and protein turnover aspect. The authors compared plasma with cartilage, bone and skin which contain a diverse set of distinct cell populations. In addition, labelled proteins from plasma are derived from the liver, which represents a proliferating tissue. Similarly, skin is also a proliferating tissue, while cartilage and bone tissue are most likely post-mitotic. It is not surprising that proliferating tissues have higher Lys6 incorporation compared to non-dividing cells. Unfortunately, the authors did not address this issue, and the comparison unfortunately remains unclear.

2. A more focused analysis of a particular cell type and its Lys6 incorporation rate at different time points or disease states is a viable alternative approach, which would also be highly informative, if not more so for selected questions. In light of the comment above, a more complete discussion of the limitations of this work is needed. It would be worth it to add a justification of why the methods chosen where chosen over other strategies to address the main thesis of the paper.

3. Overall, the statistical power is difficult to estimate and the authors should carefully consider the statistical analysis of a non-normal distributed data set. The comparison of young animals with the older stages simply represents growing and dividing tissues, which is naturally more labelled than "mature" tissue.

4. Throughout this paper, it is stated that trabecular bone was examined. The methods for collecting this bone are spartan. It appears that the part of the tibia collected is very trabecular bone poor and is mostly cortical bone. The turnover rates of these two bone compartments are very different. Please clarify what kind of bone you had and the implications of looking at only trabecular/ only cortical/ a mix for the interpretations of these results. Considering the tissue of interest appears to be trabecular bone, how did you ensure there was no contaminating marrow. Trabeculae are very small and hard to "clean".

5. This study is only male mice. Bone turnover is higher in females. In humans, females are more likely to get Osteoarthritis than males and wound healing is slower in females than males. The implications of sex on the interpretation of the results and the limitations of this study design need to be included in the discussion.

6. The authors claim that incorporation is lower in aged animals (week 45) compared to young animals (week 15). For example, Table S2 shows an incorporation rate of 2.97 +/- 2.13 at week 45 and 3.38+/-0.98 at week 15 for the cartilage tissue and protein collagen 1a1. The ANOVA calculation showed a significant change for this candidate. Do the authors believe that a fold-change of w45/w15 = 1:1.14 is biologically relevant? The same is true for most of the proteins listed in the table. Is it possible to calculate accurate incorporation rates in % of SILAC ratios of H/L = 0.01? This corresponds to an H/L ratio of 1:100 in the MS spectra. It would be helpful to show a real MS spectrum of such incorporation rates at least in the SI data. This would also show if the "heavy" peak is represented by a true isotopic pattern and whether this peak was selected for fragmentation. Alternatively, it might be that the "low" peak represents only a background signal. The calculation of the coefficient of variation could also help to document the variability of the data set. A more complete comparison of this work to similar previous studies in the field would provide both context for these results and validation of the expected ranges of turnover.

7. The median lifespan of C57BL/6J male mice is ~128 weeks. The growth plate of mice does not fuse and the femur will increase in length past 26 weeks of age and periosteal expansion is still happening at 12 months (PMID: 13678781). A sound rational for using mice under one year of age is provided. But please be careful when describing these mice. The oldest group are at best mature adult male mice and are certainly not "late adulthood" (line 229-230). The middle group are far from finished growing.

8. Could it be that older animals move less and eat less? This would also explain a slightly reduced Lys6 incorporation. Did the authors observe any possible weight gain or loss during feeding? Did the animals always take-up the same amount of Lys6 during the experiment?

9. There is a large difference between wound healing and normal skin maintenance/turnover. This this distinction is often blurred when discussing the results for skin and the implications of the results for skin.

10. Line 329-332. Mice very much lose trabecular bone without losing estrogen (PMID: 17488199) and naturally occurs without immobilization. Please reframe this section reflecting the reality of you model.

11. It might be better to focus the analysis on a specific tissue/cell type that is either proliferating or post-mitotic. Why did the authors not calculate absolute turnover rates? This would be a better comparison and could help to better assess the statistical power of their method.

12. Introduction – for completeness, please briefly also describe the use of deuterium labelling to measure proteome dynamics in tissues:

https://pubmed.ncbi.nlm.nih.gov/22915825/, https://pubmed.ncbi.nlm.nih.gov/32393437/ and https://pubmed.ncbi.nlm.nih.gov/32403418/

13. It is strongly suggested that the authors establishing if there were any alterations in relative protein abundance between age groups assessed. This will allow identification of any proteins that show both dysregulated turnover and abundance with ageing and are therefore most likely involved in age-related diseases.

14. Have the authors using Ingenuity Pathway Analysis (or similar software) to identify upstream regulators predicted to result in the differential protein synthesis observed with ageing? This could provide potential targets for age-related diseases that could be investigated further in future studies. The detailed bioinfomatic STRING analysis is certainly helpful but many points raised in the discussion are unfortunately very speculative and should be verified by experimental work. All reviewers were in agreement that a more comprehensive bioinformatics data analysis of some type is mandatory.

15. Is it possible to calculate and report individual protein half-lives/fractional synthesis rates from the data? This would enable more direct comparison of half-lives measured in previous studies.

16. Materials and methods – different extraction protocols were used for skin, compared to cartilage and bone samples. Please provide a rationale for this. Is this likely to have had an impact of the proteins identified? If so, this needs to be more completely mentioned in the discussion as a limitation of this work.

17. Results, Ln 160, Figures 3 and 4. It is apparent that incorporation rates decline with ageing across tissues, with most obvious differences between immature and mature animals and much smaller changes between mature and old animals. Are you able to present statistical analysis of these results on the graphs to identify significant differences with ageing?

18. Although the authors achieved a dilution of the label by a label switch to Ly0, this aspect was unfortunately only very little evaluate. The reviewers felt that this point should be elaborated on.

19. Protein turnover is always regulated by synthesis and degradation. The labelling of living animals with stable isotopes is a useful technique, but one should be very careful which tissues one compares and whether the incorporation rates are regulated by instrinsic protein stability or by cell division.

[Editors' note: further revisions were suggested prior to acceptance, as described below.]

Thank you for resubmitting your work entitled "Age-dependent changes in protein incorporation into collagen-rich tissues of mice by in vivo pulsed SILAC labelling" for further consideration by *eLife*. Your revised article has been evaluated by 3 peer reviewers, one of whom is a member of our Board of Reviewing Editors, and the evaluation has been overseen by Matt Kaeberlein as the Senior Editor. The reviewers have opted to remain anonymous.

All reviewers agreed that this manuscript is substantially clearer and very much improved after this extensive revision. Some matters however remain, and it was agreed after much discussion among the reviewers that an additional revision would be requested

1. The figure provided for A6 would be of benefit as a supplemental figure. However, in its current form it is not usable as the font size is impossible to read.

2. The largest concern from the reviewers was that the study remains descriptive and the underlying experiments comparing Lys-6 incorporation rates between different tissues or developmental stages remain questionable. While the authors have addressed this to some degree in the discussion (ln 457), further comment on the specific effect of cell division on Lys-6 incorporation would more clearly highlight the limitations of the study. The reviewers would have liked to see an integration of the absolute half-lives as a function of proliferation. Thus, the comparison always remains dependent on cell division, which unfortunately prevents a clear statement of the absolute protein half-life or protein turnover. Therefore, the wording "turnover" should not be used in the manuscript at all, but rather Lys6 incorporation rates (i.e. Figure 2A) in proliferating or postmitotic cells. Lys6 incorporation rates do not necessarily reflect the turnover of a protein and hence this term, while used sparingly, should be avoided overall.

---

## [Author Response]

Essential revisions:1. A major weakness is the tissue/cell turnover and protein turnover aspect. The authors compared plasma with cartilage, bone and skin which contain a diverse set of distinct cell populations. In addition, labelled proteins from plasma are derived from the liver, which represents a proliferating tissue. Similarly, skin is also a proliferating tissue, while cartilage and bone tissue are most likely post-mitotic. It is not surprising that proliferating tissues have higher Lys6 incorporation compared to non-dividing cells. Unfortunately, the authors did not address this issue, and the comparison unfortunately remains unclear.

Thank you for raising this important point which we failed to highlight in our manuscript. Yes, it is perfectly true that changes that we observe may well be related to the mitotic status of the cells perhaps commensurate with the renewable nature of each tissue. One might imagine that tissues that continue to turn over during life, such as the skin, will be synthetically more active and this will be reflected in the results. In fact, the total number of proteins incorporating Lys6 (i.e. number of proteins with a measured H/L ratio) across the three mature connective tissues are broadly the same (skin 450, bone 550 and cartilage 599 proteins, also Figure 3) and their range of label incorporation also similar (20-90% in plasma, 10-90% in skin, 2-90% in bone and 2-80% in cartilage), so our results do not appear to be biased by large differences in the global synthetic rates across tissues. Rather, it appears that the difference is in select groups of proteins only.

It is, of course, difficult to be definitive about how important this is when considering complex and simple tissues as the tissue analyses do not specifically indicate which cells are driving the observed phenotype. In the case of cartilage, we can be fairly confident that most (possibly all) cells are post-mitotic chondrocytes. In bone the cells will be a mixture of post mitotic (osteocytes), renewable (blood derived osteoclasts) and proliferating (osteoblasts) cells (although the latter are thought to fail with age). It is perhaps a surprise that post mitotic tissues like cartilage manage to maintain this level of synthesis at all. The post-mitotic tissues have striking effects in select protein groups suggesting that these cells do have a restricted synthetic lifespan. Our thesis is that this possibly explains age-related disease risk.

Action (A) 1. We have inserted a paragraph into the introduction (lines 62-67) and further commented on this in the discussion (lines 539-542). In the limitations section we also mention that we are comparing tissues that have a disparate number and type of cells (line 791). We have also substituted old Figure S2 with a new figure (Figure 3) that shows the incorporation levels between tissues and over time more clearly, and also shows the position of key matrisomal proteins.

2. A more focused analysis of a particular cell type and its Lys6 incorporation rate at different time points or disease states is a viable alternative approach, which would also be highly informative, if not more so for selected questions. In light of the comment above, a more complete discussion of the limitations of this work is needed. It would be worth it to add a justification of why the methods chosen where chosen over other strategies to address the main thesis of the paper.

We agree that understanding which cell types are driving the ageing phenotype across different tissues would be interesting and valuable. However, this type of analysis is severely limited in collagen rich tissues (in particular) as the cell behaviour is greatly influenced by the native tissue environment i.e. extracellular matrix. For example when articular chondrocytes are isolated they rapidly de-differentiate, start to proliferate and are highly synthetic. Therefore, one of the significant advantages of using this approach is that cellular activities can be examined in situ. It might be possible to combine our proteomic analysis with spatial proteomics/transcriptomics or single nuclear RNA sequencing, but this is technically very challenging in murine connective tissues and has not yet been optimised by our group. With regards to disease states, these studies are ongoing for osteoarthritis in our group and will follow, but we feel detract from the main messages of the current paper.

We have incorporated a comment to stress the advantage of being able to analyse synthetic activity in complex tissues without disrupting the cells and matrix in the introduction (line 62 and lines 67-68).

3. Overall, the statistical power is difficult to estimate and the authors should carefully consider the statistical analysis of a non-normal distributed data set. The comparison of young animals with the older stages simply represents growing and dividing tissues, which is naturally more labelled than "mature" tissue.

Our ‘within group’ datasets where we examined the normality of each individual protein against the Shapiro-Wilk normality test and for variance against the Levene test demonstrated that >80% of proteins were normally distributed. Therefore for protein:protein comparisons we applied Student t-tests for pairwise comparisons. We performed group wise comparisons between young and older adults using non-parametric tests.

We agree that the differences observed between skeletally immature and mature groups likely represents growing tissues. These data (presented in Figures 2-6) were intended to demonstrate that the in vivo SILAC methodology was fit for purpose i.e. to demonstrate a proof of concept before going on to look in more detail at the differences in skeletally mature tissues with age. In addition we also feel that these data have intrinsic interest. For instance we were able to define a number of matrisomal proteins that appear to switch off after skeletal maturity e.g. keratocan in all three tissues, as well as those that switch on in adult tissues and are not apparently synthesised during growth e.g. type XV collagen in bone and versican in articular cartilage.

We have added detail to the statistical methods section to clarify the statistical tests used (lines 941-948). We have added in the results that whilst most matrisomal proteins go down after skeletal maturity, some proteins are incorporated for the first time in adult tissues (lines 392-393).

4. Throughout this paper, it is stated that trabecular bone was examined. The methods for collecting this bone are spartan. It appears that the part of the tibia collected is very trabecular bone poor and is mostly cortical bone. The turnover rates of these two bone compartments are very different. Please clarify what kind of bone you had and the implications of looking at only trabecular/ only cortical/ a mix for the interpretations of these results. Considering the tissue of interest appears to be trabecular bone, how did you ensure there was no contaminating marrow. Trabeculae are very small and hard to "clean".

Thank you for pointing out this error. The bone that we collected was mainly cortical bone, not trabecular. We chose this region firstly because we wanted to avoid parts of the bone that were near cartilage/cartilaginous tissue such as subchondral bone and the growth plate, and also because it had clear tissue landmarks, which enabled consistent tissue collection in each animal irrespective of age. The diaphyseal area of tibia from which bone samples were taken has a high cortex-to-trabecular bone ratio. In humans, trabecular bone has a higher turnover rate than cortical bone and it is possible that changes in the ratio of cortical to trabecular bone with age could have influenced our results.

The bone marrow was washed through with PBS injected with a fine needle and syringe and washes were checked to ensure that they were acellular using a stereo microscope. It is correct that we cannot guarantee 100% marrow-free bone.

We have corrected reference to trabecular bone throughout the text and have added a sentence in the limitations section of the discussion to highlight the potential change of cortical to trabecular bone ratio with age that may have influenced the results (lines 731-732). Bone preparation has been clarified in the methods (lines 854-856).

5. This study is only male mice. Bone turnover is higher in females. In humans, females are more likely to get Osteoarthritis than males and wound healing is slower in females than males. The implications of sex on the interpretation of the results and the limitations of this study design need to be included in the discussion.

We agree, which is why we controlled for gender. Time and funding allowing we would have liked to do this in age-matched female mice also. We have added this as a limitation in discussion (737-788).

6. The authors claim that incorporation is lower in aged animals (week 45) compared to young animals (week 15). For example, Table S2 shows an incorporation rate of 2.97 +/- 2.13 at week 45 and 3.38+/-0.98 at week 15 for the cartilage tissue and protein collagen 1a1. The ANOVA calculation showed a significant change for this candidate. Do the authors believe that a fold-change of w45/w15 = 1:1.14 is biologically relevant? The same is true for most of the proteins listed in the table. Is it possible to calculate accurate incorporation rates in % of SILAC ratios of H/L = 0.01? This corresponds to an H/L ratio of 1:100 in the MS spectra. It would be helpful to show a real MS spectrum of such incorporation rates at least in the SI data. This would also show if the "heavy" peak is represented by a true isotopic pattern and whether this peak was selected for fragmentation. Alternatively, it might be that the "low" peak represents only a background signal. The calculation of the coefficient of variation could also help to document the variability of the data set. A more complete comparison of this work to similar previous studies in the field would provide both context for these results and validation of the expected ranges of turnover.

Thank you for the comment. We are not in a position to claim biological, nor indeed, clinical relevance from individual protein results, but think that percentages of new protein incorporation are valuable when interpreted as a trend of incorporation rates across protein groups. Even though we will always be limited by the sensitivity of protein incorporation rates, we can claim some confidence by showing that we were able to measure a wide range of H/L ratios for a diverse range of proteins and time points in all tissues and in a reproducible fashion between replicates.

We include an example spectra of two col1a1 peptides showing distinct peaks in labelled and unlabelled peptide for the 15 and 45 week data. We would be happy to include this as an extra supplementary figure if felt necessary. (Editors note: this has now been added as Figure 4—figure supplement 1).

7. The median lifespan of C57BL/6J male mice is ~128 weeks. The growth plate of mice does not fuse and the femur will increase in length past 26 weeks of age and periosteal expansion is still happening at 12 months (PMID: 13678781). A sound rational for using mice under one year of age is provided. But please be careful when describing these mice. The oldest group are at best mature adult male mice and are certainly not "late adulthood" (line 229-230). The middle group are far from finished growing.

We have now replaced the terms to read “young adult” and “older adult”. We avoided “mature adult” as we felt that this was confusing as we use the term skeletal maturity for both groups C and D.

8. Could it be that older animals move less and eat less? This would also explain a slightly reduced Lys6 incorporation. Did the authors observe any possible weight gain or loss during feeding? Did the animals always take-up the same amount of Lys6 during the experiment?

We recorded animal weight throughout the experiment (see Author response image 1). Mice gained weight in the same way that they do when feeding on standard diet (based on reference growth curves). We cannot exclude that older adult mice moved less as we didn’t formally assess their movement. This might be expected from other ageing studies and could contribute to the changes we are seeing. We didn’t observe any obvious differences in intake between young and older adults although this wasn’t measured formally.

**Author response image 1. sa2fig1:** Weight gain for each of the experimental mice over the period of SILAC heavy diet (3-10 weeks, 11-15 weeks or 42-45 weeks). These weight curves are consistent with reference growth curves in C57BL/6 mice.

We have added a comment about gaining expected weight in the methods section (lines 842-843).

9. There is a large difference between wound healing and normal skin maintenance/turnover. This this distinction is often blurred when discussing the results for skin and the implications of the results for skin.

We recognise this in our efforts to extrapolate the data to an age-related clinical problem in skin. We have added a comment about not necessarily being able to extrapolate to wounding as this was not a wounding study (lines 725-726).

10. Line 329-332. Mice very much lose trabecular bone without losing estrogen (PMID: 17488199) and naturally occurs without immobilization. Please reframe this section reflecting the reality of you model.

Thank you. We have clarified this in the text. See changes (lines 736-737).

11. It might be better to focus the analysis on a specific tissue/cell type that is either proliferating or post-mitotic. Why did the authors not calculate absolute turnover rates? This would be a better comparison and could help to better assess the statistical power of their method.

One of the primary purposes for doing this project was to ask whether change in synthetic rates alter between different collagen rich tissues and whether this could account for very poor repair responses in select tissues that predispose to age-related disease (e.g. cartilage and osteoarthritis). This question is not the same as asking about individual turnover rates of proteins within tissues (reflecting the balance between synthesis and degradation), which is what the field has largely focused on using other methodologies. In order to address protein turnover rates we would have needed to do a second experimental group after a 3 week washout (similar to comparison of group A and B in Figure 2). We hope we have stressed this now in the substantially revised manuscript.

12. Introduction – for completeness, please briefly also describe the use of deuterium labelling to measure proteome dynamics in tissues:

https://pubmed.ncbi.nlm.nih.gov/22915825/, https://pubmed.ncbi.nlm.nih.gov/32393437/ and https://pubmed.ncbi.nlm.nih.gov/32403418/

This has now been added to introduction (lines 202-207).

13. It is strongly suggested that the authors establishing if there were any alterations in relative protein abundance between age groups assessed. This will allow identification of any proteins that show both dysregulated turnover and abundance with ageing and are therefore most likely involved in age-related diseases.

This is an important consideration. We therefore re-examined raw data and used total iBAQ values to calculate an abundance score for each protein. Interestingly, very few proteins showed regulated abundance levels when considering total protein levels. Articular cartilage was the only tissue where a few proteins were regulated significantly after multiple correction (five out of six of these were also regulated in the labelled analysis). These results most likely reflect the enhanced sensitivity that the SILAC labelling method affords, although clearly this could also be related to other factors that contribute to protein turnover.

We have added a new supplementary figure showing raw data correlation and iBAQ volcano plots and have mentioned these briefly in the results (lines 429-433).

14. Have the authors using Ingenuity Pathway Analysis (or similar software) to identify upstream regulators predicted to result in the differential protein synthesis observed with ageing? This could provide potential targets for age-related diseases that could be investigated further in future studies. The detailed bioinfomatic STRING analysis is certainly helpful but many points raised in the discussion are unfortunately very speculative and should be verified by experimental work. All reviewers were in agreement that a more comprehensive bioinformatics data analysis of some type is mandatory.

We did consider this type of pathway analysis but felt that we would likely be underpowered to draw robust conclusions (these analyses being best validated for transcriptomic datasets that are usually substantially bigger). However, in response to reviewers’ suggestion we have performed pathway enrichment using various proprietary software, including IPA and DAVID. Cluster enrichment using DAVID showed similar results to STRING. Using IPA, several pathways were identified although none reaching statistical significance after correction.

We have included the IPA pathway enrichment analysis as supplementary information (Figure 7—figure supplementary 3). The results are mentioned in results (lines 472-476) and discussion (559-560).

15. Is it possible to calculate and report individual protein half-lives/fractional synthesis rates from the data? This would enable more direct comparison of half-lives measured in previous studies.

Unfortunately, we are unable to calculate individual protein half-lives from single time points. As mentioned above, our focus was more on the synthetic activity of the tissue rather than the turnover of individual proteins, although we accept that having this information would have helped to validate results from similar studies using different methodologies.

16. Materials and methods – different extraction protocols were used for skin, compared to cartilage and bone samples. Please provide a rationale for this. Is this likely to have had an impact of the proteins identified? If so, this needs to be more completely mentioned in the discussion as a limitation of this work.

Different connective tissues often require different protocols to solubilise the tissue due to variable degrees of collagen cross-linking, mineralisation etc. For each tissue we established protocols to solubilise the tissue prior to doing the labelling study. This is why they have different protocols. Articular cartilage solubilisation was, as expected, most challenging, presumed to be due to accumulation of long-lived cross-linked fibrillar collagens. It is likely that this will underestimate the calculated % collagen that is labelled but it is unlikely to account for the % labels of other proteins as these are generally soluble under such conditions.

This limitation is discussed in the limitation section of the discussion (line 728).

17. Results, Ln 160, Figures 3 and 4. It is apparent that incorporation rates decline with ageing across tissues, with most obvious differences between immature and mature animals and much smaller changes between mature and old animals. Are you able to present statistical analysis of these results on the graphs to identify significant differences with ageing?

The statistical significant differences are presented in new figure 4 and 5, with explanation to code in the figure legends. The information can also be found in Figure 4-source data and Figure 5-source data.

18. Although the authors achieved a dilution of the label by a label switch to Ly0, this aspect was unfortunately only very little evaluate. The reviewers felt that this point should be elaborated on.

We accept this criticism. When designing these experiments we did the first part of this paper (groups A and B in skeletally immature animals) as a proof of concept that in vivo SILAC would provide a valid robust measure of protein synthesis in tissues in vivo. We agree that these results deserve more attention, so have (conscious of space restraints) added further discussion around these results.

Further granularity of these results is given in results, lines 339-344.

19. Protein turnover is always regulated by synthesis and degradation. The labelling of living animals with stable isotopes is a useful technique, but one should be very careful which tissues one compares and whether the incorporation rates are regulated by instrinsic protein stability or by cell division.

We take the point that incorporation rates will be in part regulated by intrinsic protein stability (which could change with age), although are not clear how cell division affects incorporation rates except in proteins specifically implicated in replicative activities. Our data clearly show that cartilage and bone continue to synthesise proteins across the healthy lifecourse albeit that the reduction in synthetic activity appears to be more age-sensitive and selective for long-lived proteins such as collagen type II.

We stress that we are only measuring incorporation, not stability of proteins in the final paragraph of the discussion (lines 795-796). [Editors' note: further revisions were suggested prior to acceptance, as described below.]

1. The figure provided for A6 would be of benefit as a supplemental figure. However, in its current form it is not usable as the font size is impossible to read.

We have created a new supplementary figure to Figure 4 which we have spread out over 4 pages to improve clarity, with reference to this in the text lines 287 and 572-3.

2. The largest concern from the reviewers was that the study remains descriptive and the underlying experiments comparing Lys-6 incorporation rates between different tissues or developmental stages remain questionable. While the authors have addressed this to some degree in the discussion (ln 457), further comment on the specific effect of cell division on Lys-6 incorporation would more clearly highlight the limitations of the study. The reviewers would have liked to see an integration of the absolute half-lives as a function of proliferation. Thus, the comparison always remains dependent on cell division, which unfortunately prevents a clear statement of the absolute protein half-life or protein turnover. Therefore, the wording "turnover" should not be used in the manuscript at all, but rather Lys6 incorporation rates (i.e. Figure 2A) in proliferating or postmitotic cells. Lys6 incorporation rates do not necessarily reflect the turnover of a protein and hence this term, while used sparingly, should be avoided overall.

We accept that the term “turnover” is misleading when we have not measured the half-lives of individual proteins, and where we are comparing tissues that have different cell renewal rates which may also change with age. We have therefore gone through the manuscript carefully and have edited “turnover” to 13C6 Lys incorporation or equivalent in each of these places. These changes can be found on lines 34, 37, 165, 192, 485, 590, 594, 871, 1031, and are highlighted in green for ease of reading.

There are two cases where we feel the term “turnover” is still appropriate. Firstly in the introduction, where we refer to other published studies, and secondly in Figure 2 where we broadly categorise high and low turnover protein groups within an individual tissue. As mitotic activity is controlled for within each tissue at baseline and after the washout period, we believe this to be a reasonable interpretation of the data. We reinforce this distinction by adding “high turnover for a given tissue” to line 187.

Conversely, later in Figure 2 where we compare profiles between tissues we have removed the term “turnover” and substituted it for 13C6 Lys incorporation or equivalent. We hope this is acceptable.

We have further tried to clarify this by adding the following to line 315 of the discussion:

“The tissue turnover, which is likely to reflect both protein turnover and cellular renewal, can be divided into tissue modelling….”.